# Tyrosine 67 Phosphorylation Controls Respiration and Limits the Apoptotic Functions of Cytochrome *c*

**DOI:** 10.3390/cells14130951

**Published:** 2025-06-21

**Authors:** Junmei Wan, Paul T. Morse, Matthew P. Zurek, Alice A. Turner, Asmita Vaishnav, Arthur R. Salomon, Brian F. P. Edwards, Tasnim Arroum, Maik Hüttemann

**Affiliations:** 1Center for Molecular Medicine and Genetics, Wayne State University, Detroit, MI 48201, USAho0066@wayne.edu (T.A.); 2Department of Biochemistry, Microbiology and Immunology, Wayne State University, Detroit, MI 48201, USA; 3MCB Department, Brown University, Providence, RI 02903, USA

**Keywords:** cytochrome *c*, phosphorylation, cell signaling, apoptosis, heart, electron transport chain, reactive oxygen species, mitochondrial membrane potential

## Abstract

Cytochrome *c* (Cyt*c*) is a multifunctional protein, essential for respiration and intrinsic apoptosis. Post-translational modifications of Cyt*c* have been linked to physiological and pathophysiologic conditions, including cancer. Cyt*c* tyrosine 67 (Y67) is a conserved residue that is important to the structure and function of Cyt*c.* We here report the phosphorylation of Y67 of Cyt*c* purified from bovine heart mapped by mass spectrometry. We characterized the functional effects of Y67 Cyt*c* modification using in vitro and cell culture models. Y67 was mutated to the phosphomimetic glutamate (Y67E) and to phenylalanyl (Y67F) as a control. The phosphomimetic Y67E Cyt*c* inhibited cytochrome *c* oxidase (COX) activity, redirecting energy metabolism toward glycolysis, and decreased the pro-apoptotic capabilities of Cyt*c*. The phosphomimetic Y67E Cyt*c* showed a significantly impaired rate of superoxide scavenging and a reduced rate of oxidation by hydrogen peroxide*,* suggesting a lower ability to transfer electrons and scavenge reactive oxygen species (ROS). Phosphomimetic Y67E replacement led to an almost complete loss of cardiolipin peroxidase activity, pointing to a central role of Y67 for this catalytic function of Cyt*c*. In intact cells, phosphomimetic replacement leads to a reduction in cell respiration, mitochondrial membrane potential, and ROS levels. We propose that Y67 phosphorylation is cardioprotective and promotes cell survival.

## 1. Introduction

Cytochrome *c* (Cyt*c*) is an essential small globular protein present in the mitochondrial intermembrane space and is linked with a heme group through thioether bonds with cysteine residues 14 and 17. The iron in the center of the heme is in a hexacoordinate configuration with H18 and M80 as axial ligands. In the oxidized state, the heme iron–M80 bond contributes to the weak 695 nm absorption band in the spectrum of Cyt*c*. The heme group surrounded by aliphatic and aromatic amino acid side chains creates a hydrophobic environment, which contributes to the high redox potential of Cyt*c* of about 260 mV [1,2].

Cyt*c* functions as an electron carrier between *bc*_1_ complex (complex III) and cytochrome *c* oxidase (complex IV, COX) in the electron transport chain (ETC) embedded in the inner mitochondrial membrane, resulting in COX reducing O_2_ to form H_2_O. As electrons flow through the ETC, complexes I, III, and IV pump protons from the matrix into the intermembrane space. This proton gradient is harnessed by ATP synthase (complex V) to produce ATP. Given that the transfer of electrons from Cyt*c* to COX to O_2_ is the proposed rate-limiting step of the ETC in intact cells [3,4], the integrity of Cyt*c* structure and function is very important for the regulation of mitochondrial respiration. Not only is it crucial for cell life, Cyt*c* also serves as a trigger of intrinsic apoptosis when released from the mitochondria into the cytosol during cellular stress. Cyt*c* interacts with the protein apoptotic protease activating factor 1 (Apaf-1) to form the apoptosome in the presence of ATP, which leads to the recruitment and activation of caspase-9 that in turn cleaves and activates downstream caspase-3, leading to cell death [1].

In addition, Cyt*c* exerts multiple other functions such as acting as a reactive oxygen species (ROS) scavenger; a cardiolipin peroxidase; an electron acceptor of the Erv1-Mia40 protein import pathway by which nuclear-encoded proteins are imported to the mitochondrial intermembrane space; a ROS producer via reduction of p66^shc^; a regulator in DNA damage response; and others [1,5,6,7]. In the mitochondrial matrix, superoxide dismutase detoxifies ROS. However, Cyt*c* also plays a key role in the detoxification of ROS within the intermembrane space. Oxidized Cyt*c* accepts unpaired electrons from superoxide and transfers them to COX for energy production [8,9] or reduced Cyt*c* donates electrons to hydrogen peroxide (H_2_O_2_) to generate H_2_O [10]. Therefore, Cyt*c* plays a vital role in maintaining redox homeostasis during respiration.

Cyt*c* catalyzes the peroxidation of cardiolipin (CL), a mitochondria-specific lipid localized in the inner mitochondrial membrane [11]. About 15–20% of Cyt*c* is bound to CL. Cyt*c* catalyzes the peroxidation of CL when under conditions of high oxidative stress [11,12]. The interaction of Cyt*c* with CL, which is enhanced at low ATP levels [13], results in partial protein unfolding and conformational changes in the M80–heme iron coordination, spectral changes in the heme iron spin state, and increased peroxidase activity [14,15]. During apoptosis, additional CL from the inner leaflet of the inner mitochondrial membrane redistributes to the outer leaflet and becomes available for peroxidation [16]. After CL peroxidation Cyt*c* can be more easily released from the mitochondria due to the decreased binding affinity following the oxidation of CL [16] or the autocatalytic carbonylation of Cyt*c* [17]. Recently, Cyt*c* was shown to translocate to the nucleus where it interacts with nucleophosmin [18], and to modulate chromatin remodeling by interacting with the oncoprotein SET/TAF-Iβ in the nucleus after its release from the mitochondria [19,20], which is regulated by tyrosine phosphorylation of Cyt*c* [21].

Given the crucial functions of Cyt*c*, the protein is strictly regulated in several ways: allosteric regulation by ATP, tissue-specific isoforms (somatic and testes), and post-translational modifications (reviewed in [22,23]). Cyt*c* has been shown to undergo a wide variety of post-translational modifications, including acetylation, carbonylation [17,24,25], deamidation [26], glycation [27,28,29,30], glycosylation [31,32], homocysteinylation [33,34,35,36], nitration [37,38,39,40,41], nitrosylation [42], phosphorylation, and sulfoxidation [43,44,45,46]. We previously reported that Cyt*c* can be phosphorylated or acetylated in vivo, which is of relevance to both physiological and pathological conditions. Phosphorylation or acetylation at different sites of Cyt*c* impact the biochemical properties and multiple functions of the protein, thus affecting its role in mitochondrial respiration and apoptosis. Interestingly, the phosphorylations appear to be tissue-specific and are present under basal conditions: T28 [47,48] and T58 [49] in the kidney, S47 in the brain [48,50,51], Y48 in the liver [21,52,53,54,55], and T49 [56] and Y97 in the heart [54,57,58,59]. In contrast, the acetylations appear to be disease-specific: acetylation of K39 occurs during ischemia in skeletal muscle, which promotes the utilization of oxygen and protects skeletal muscle from reperfusion injury [60], while acetylation of K53 occurs in prostate cancer, resulting in Warburg metabolism and evasion of apoptosis [61,62,63]. Recently, Cyt*c* was shown to interact with the E3 ligase, Pirh2, in the cytosol after being released from the mitochondria during apoptosis in an N2A cell culture model of Alzheimer’s disease [64]. The authors further showed that Cyt*c* can be ubiquitinated by Prih2 in vitro which should be further confirmed by in vivo studies.

The purpose of the current study was to expand our understanding of the regulation of Cyt*c* by mapping and functionally studying a new phosphorylation site, Y67 of Cyt*c*, and to test the hypothesis if this novel phosphorylation leads to similar beneficial functional effects as previously reported for the other identified phosphorylation sites. We propose that Cyt*c* phosphorylation under normal conditions plays a central role in regulating respiration and apoptosis by maintaining intermediate mitochondrial membrane potential (∆Ѱ_m_) within the physiological range of 80–140 mV. This allows for near-maximal ATP production and at the same time minimizes ROS, which are produced exponentially at pathologically high ∆Ѱ_m_ levels exceeding 140 mV [5,65,66]. Typically, these phosphorylations, which are present under basal conditions, are lost during cellular stress such as ischemia, which then sensitizes the tissue to reperfusion injury. So far, Cyt*c* phosphorylation has been linked to AMPK and the insulin and Akt signaling pathways [47,51,67].

We earlier reported that bovine heart Cyt*c* is phosphorylated on Y97 [57]. Here we mapped a novel second phosphorylation site mapped to Y67 of Cyt*c* purified from bovine heart and report its in vitro biochemical characterization through phosphomimetic replacement. We show that replacing tyrosine with glutamate (Y67E) inhibits the central functions of Cyt*c* related to respiration and apoptosis in support of our proposed model that Cyt*c* phosphorylation controls respiration to prevent ∆Ѱ_m_ hyperpolarization and subsequent excessive ROS generation that causes apoptosis.

## 2. Methods

All chemicals and reagents were purchased from MilliporeSigma (Burlington, MA, USA) unless otherwise specified.

### 2.1. Isolation of Cytc from Bovine Heart Tissue

Cow hearts were obtained from a slaughterhouse (Dunbar Meats; Milan, MI, USA). Acquiring approval for the use of discarded slaughterhouse animal tissues is exempt from review by Wayne State University’s Institutional Animal Care and Use Committee. Hearts from freshly slaughtered cows were snap-frozen on dry ice and stored at −80 °C until used for Cyt*c* purification [52,57]. Homogenized heart tissues were acid-extracted overnight in 100 mM phosphate buffer at pH 4.5. Homogenates were centrifuged at 15,810× *g* for 35 min at 4 °C. The pH of the supernatant was adjusted to 7.4 using KOH. A protease inhibitor (1 mM phenylmethylsulfonyl fluoride) and phosphatase inhibitors (10 mM KF and 1 mM activated sodium orthovanadate) were added. This solution was then centrifuged once again at 15,810× *g* for 35 min at 4 °C. The resulting supernatant was adjusted to the conditions (pH 7.4, conductivity of 3.6 mS/cm, at 4 °C) of a DE52 anion exchange column (Whatman; Piscataway, NJ, USA) and loaded onto the column. Flow-through from the DE52 column was adjusted to the conditions (pH 6.5, conductivity 5.5 mS/cm, at 4 °C) of a CM52 cation exchange column (Whatman) and loaded onto the column. Cyt*c* was oxidized with 2 mM potassium ferricyanide and eluted using a stepwise gradient of 30–150 mM phosphate buffers at pH 6.5. Concentration via vacuum centrifugation and desalting were performed followed by storage at −80 °C.

### 2.2. Mass Spectrometry of Purified Cytc to Detect Site-Specific Phosphorylation

Phosphorylation site mapping on purified heart Cytc was performed as previously described [57]. Tryptic digestion and the enrichment of phosphopeptides with titanium dioxide were performed, followed by desalting with Sep-Pak C18 reversed-phase chromatography as described in [7], and drying in a Speed Vac plus (Thermo Savant, Holbrook, NY, USA). Electrospray ionization was used and MS/MS spectra were acquired (LTQ Orbitrap-Velos, Thermo Scientific, Waltham, MA, USA). Peptide sequences from the UniProt protein database were assigned and searched with the MASCOT algorithm for posttranslational modifications. Phosphopeptide spectra were manually verified.

### 2.3. Mutagenesis, Expression, and Purification of Recombinant Cytc

Rat somatic Cyt*c* cDNA was cloned into the pLW01 expression vector (a gift from Dr. Lucy Waskell, University of Michigan) [53,68], which also contains the cDNA encoding *S. cerevisiae* heme lyase (CYC3). The codon corresponding to Y67 of the somatic rodent Cyt*c* cDNA cloned into the PLW01 expression vector was mutated to a negatively charged glutamate residue (Y67E) as a phosphomimetic replacement, as well as a phenylalanine residue (Y67F) as an additional neutral and unphosphorylatable control. Mutants were generated using the QuickChange lightning site-directed mutagenesis kit (Agilent; Santa Clara, CA, USA) according to the manufacturer’s protocol via the following primers: Y67F forward primer 5′-CCCTGATGGAGTTTTTGGAAAATCC-3′ (T_m_ = 58 °C), Y67F reverse primer 5′-GGATTTTCCAAAAACTCCATCAGGG-3′ (T_m_ = 58 °C), Y67E forward primer 5′-CCCTGATGGAGGAGTTGGAAAATCC-3′ (T_m_ = 61 °C), and Y67E reverse primer 5′-GGATTTTCCAACTCCTCCATCAGGG-3′ (T_m_ = 61 °C) [31]. Parental DNA was digested by the DpnI restriction enzyme. The mutated constructs were transformed with XL10-Gold Ultracompetent cells (Stratagene Technologies; La Jolla, CA, USA). Plasmids were purified from individual colonies using the Wizard Plus SV miniprep purification system (Promega; Madison, WI, USA). The presence of the desired mutations was confirmed using DNA sequencing (Genewiz; South Plainfield, NJ, USA). Phosphomimetic glutamate mutation can functionally mimic the phosphorylated protein [53]. *E. coli* C41 (DE3) cells (Lucigen; Middleton, WI, USA) were transformed via heat shock [53]. Transformed bacteria were initially cultured in 10 mL of LB broth medium supplemented with 0.1 mg/mL ampicillin and allowed to grow at 37 °C overnight while shaking. These starter cultures were then inoculated in multiple 1 L flasks of TB medium (Difco, BD; Franklin Lakes, NJ, USA) with 0.1 mg/mL carbenicillin and allowed to grow for approximately 5 h until A_600nm_ reached a 0.8–1.2 optical density. The expression of Cyt*c* was then induced by 100 µM isopropyl β-D-1-thiogalactopyranoside, and the protein was overexpressed in the culture at 37 °C for 2–6 h, which was determined by the presence of the characteristic pink color of Cyt*c* observed when 1 mL of media was pelleted. Cells were harvested by centrifugation at 8400× *g*, 4 °C, for 40 min. The bacterial pellets were resuspended in 20 mM phosphate buffer, pH 7.4, supplemented with a protease inhibitor (0.2 mM phenylmethylsulfonyl fluoride). Approximately 40 mL of the resuspended cells was sonicated at a time using a Branson Sonifier W-350 cell disruptor (Branson Sonic Power Co.; Danbury, CT, USA) at output control 5 until the total mixture was sonicated twice. The cells were further lysed using an SLM Aminco French pressure cell press (American Instrument Co.; Silver Spring, MD, USA). The lysates were centrifuged at 20,000× *g*, 4 °C, for 45 min, and the resulting supernatants were adjusted to pH 7.4. Cyt*c* variants were purified by ion exchange chromatography, as described above, followed by size exclusion using a Sephacryl S-100 (GE Healthcare; Chicago, IL, USA) gel filtration column. The resulting protein solutions were then further concentrated and buffer-exchanged to water using Amicon Ultra-15 10 kDa centrifugal filter units (MilliporeSigma; Burglington, MA, USA). Protein concentration was determined spectrophotometrically, while protein purity was determined using Coomassie blue staining with a 10% tris-tricine SDS-PAGE gel [47].

### 2.4. Concentration Determination of Cytc

Purified Cyt*c* was completely reduced with an excess of sodium dithionite (100 mM) or fully oxidized with 100 mM potassium ferricyanide and desalted via an NAP-5 size exclusion column (GE Healthcare; Chicago, IL, USA). The absorbance at 550 nm was recorded using a Jasco V-570 double-beam spectrophotometer (JASCO Corporation; Hachioji, Tokyo, Japan) using a 2 nm bandwidth. The concentration was calculated using the formula [Cyt*c*] in mM = (Abs550_reduced_ − Abs550_oxidized_)/(19.6 mM/cm × 1 cm) × dilution factor.

### 2.5. Measurement of Cytochrome c Oxidase Activity

Previously isolated regulatory-competent pig heart COX was used for the reaction with Cyt*c* [69]. To remove bound cholate and damaged cardiolipin, COX was dialyzed overnight at 4 °C in the presence of cardiolipin and ATP, as described in [58]. COX activity was measured using 200 nM COX via an oxygen electrode (Oxygraph system, Hansatech; Pentney, UK) at 25 °C in 220 µL of COX measuring buffer (10 mM K-HEPES, pH 7.4, 40 mM KCl, 1% Tween 20) and 20 mM ascorbate as a Cyt*c* reductant. Cyt*c* variants were titrated (0–15 μM) into the measuring chamber, and oxygen consumption was measured and analyzed using the Oxygraph software (version 1.0.48). COX activity is reported as a turnover number (sec^−1^).

### 2.6. Caspase-3 Activity

Caspase-3 activation was assayed using an in vitro cell-free apoptosis detection system with cytosolic extracts containing Apaf-1 from Cyt*c*-/- rat embryonic cell line (ATCC^®^ CRL-2613™; Manassas, VA, USA), as previously described [53,70]. Caspase-3 activity induced by purified, recombinant Cyt*c* was measured using the EnzChek Caspase-3 assay kit (Invitrogen; Carlsbad, CA, USA). Cell extracts (1 mg/mL) were incubated with recombinant Cyt*c* variants (20 μg/mL) for 2 h at 37 °C. Fluorescence was detected using a Fluoroskan Ascent FL plate reader (Labsystems, Thermo Scientific; Waltham, MA, USA) with excitation/emission wavelengths of 485 nm/527 nm. Fluorescence values (arbitrary units) were measured every 30 min for 3 h at 20 °C. Fluorescence values from the caspase inhibitor and background readings (cytosolic extract without Cyt*c*) were subtracted from the results. Caspase-3 activity is reported as a percentage of change compared to the WT.

### 2.7. Measurement of Cytc Redox Potential

The midpoint redox potential (E°’) was determined spectrophotometrically using the equilibration method as previously described [71] with 2,6-dichloroindophenol (DCIP, E°’ = 237 mV) as a reference compound. One mL of Cyt*c* solution (2 mg/mL) was mixed in a spectrophotometric cuvette with 2 mL of 50 mM citrate buffer, pH 6.5, 100 μL of 1 mM DCIP, and 25 μL of 1 mM K_3_Fe(CN)_6_ to fully oxidize Cyt*c*. Absorbance values corresponding to fully oxidized Cyt*c* (A_550_ − A_570_) and DCIP (A_600_) were recorded using a Jasco V-570 double-beam spectrophotometer. The mixture was then sequentially reduced by titrating 5 mM ascorbate, pH 6.5, in 1 μL increments, and absorbance values were acquired at each step. Data were plotted as log(DCIP_ox_/DCIP_red_) versus log(Cyt*c*_ox_/Cyt*c*_red_), which yielded a linear line with a slope, *n*_DCIP_/*n*_Cyt*c*_, and a y-intercept, *n*_Cyt*c*_/59.2 (E_Cyt*c*_ − E_DCIP_). These values were used to calculate the E°’ for the recombinant Cyt*c* variants via the Nernst equation [53,71]. E°’ is reported as mV.

### 2.8. Measurement of Cytc Rate of Oxidation

The rates of oxidation of the Cyt*c* variants with H_2_O_2_ were measured spectrophotometrically at 550 nm as previously described [47,72]. Cyt*c* variants were fully reduced with sodium dithionite and subsequently desalted by passing through NAP5 columns (GE Healthcare; Piscataway, NJ, USA). Ferro-Cyt*c* variants (15 μM) were prepared in 0.2 M Tris-Cl, pH 7.0, and a baseline absorbance at 550 nm was recorded. The oxidation reaction was initiated with 100 µM H_2_O_2_. The decrease in absorbance at 550 nm was then tracked every 10 s for 2 min, which corresponds to the oxidation of Cyt*c*. The initial rate of oxidation is reported in μM/s.

### 2.9. Measurement of Cytc Rate of Reduction by Superoxide Scavenging

The rates of reduction of the Cyt*c* variants with superoxide were measured spectrophotometrically at 550 nm as previously described [73,74]. Cyt*c* variants were fully oxidized with potassium ferricyanide and subsequently desalted by passing through NAP5 columns (GE Healthcare). Ferri-Cyt*c* variants (10 μM) were prepared in 1x PBS along with 66 μM hypoxanthine and 14.65 nM catalase, and a baseline absorbance at 550 nm was recorded. The production of superoxide was initiated with 222.2 nM xanthine oxidase. The increase in absorbance at 550 nm was then tracked every 10 s for 2 min, which corresponds to the reduction of Cytc. As a negative control, 523.2 nM superoxide dismutase was also added. The initial rate of reduction is reported in μM/s.

### 2.10. Heme Degradation Assay

The degradation of the covalently attached heme groups of the Cyt*c* variants was spectrophotometrically measured via a decrease in the Soret band at 408 nm in the presence of high concentrations of H_2_O_2_, as previously described [72]. Cyt*c* was fully oxidized with potassium ferricyanide and desalted using NAP5 columns. Ferri-Cyt*c* variants (5 μM) were prepared in 50 mM phosphate buffer, pH 6.1, and a baseline absorbance at 408 nm was measured. The degradation of the heme group was initiated with 3 mM H_2_O_2_. The decrease in absorbance at 408 nm was measured at 60, 200, 400, 600, and 800 s, which corresponds to the degradation of the heme group. The heme degradation is reported as a percentage of change in absorbance at 208 nm compared to the baseline absorbance for each Cyt*c* variant.

### 2.11. Cardiolipin Peroxidase Activity

The cardiolipin peroxidase activities of the Cyt*c* variants were measured via Amplex red, as previously described [49]. Cardiolipin/phospholipid mixtures containing 0%, 20%, 30%, and 50% of tetraoleoyl-cardiolipin with the remainder being 1,2-dioleoyl-sn-glycero-3-phosphocholine were generated in 20 mM K-HEPES buffer, pH 7.2. Lipids were reconstituted into lipid mixtures via 5 rounds of sonication for 30 s each round on ice, with one-minute breaks between sonication rounds. Lipid mixtures (25 µM) were incubated with Cyt*c* variants (1 μM) for 10 min to allow for Cyt*c*–cardiolipin binding. Baseline fluorescence was measured using a Fluoroskan Ascent FL microplate reader with excitation/emission wavelengths of 530 nm/590 nm. The peroxidase reaction was initiated with 10 μM Amplex Red and 5 μM H_2_O_2_. Cardiolipin peroxidase activity is reported as fluorescence/minute (arbitrary units/min).

### 2.12. Molecular Dynamics Simulations

Molecular dynamic simulations of Cyt*c* variants were performed using YASARA version 19.12.14 [75]. The initial Cyt*c* crystal structure was obtained from RCSB.org (ID 5c0z.pdb). Molecule A was used for simulations. Amino acid replacements were performed with Yasara for each variant in addition to wild type and phospho-tyrosine at residue 67. Energy minimizations were performed in the YASARA2 forcefield. Molecular dynamics simulations used the conservative “slow” protocol and the Amber 14 forcefield [76]. Data are plotted as the average RMSF of all atoms for each amino acid residue within 100 nanosecond blocks. Simulations were run until equilibrium was reached.

### 2.13. Establishment of Stable Cell Lines Expressing Cytc Variants

Previously generated somatic rodent Cyt*c* (WT) cloned into pBABE-puro expression plasmid (Addgene; Cambridge, MA, USA) was used to generate Y67F and Y67E Cyt*c* variants using site-directed mutagenesis (Agilent Technologies; Santa Clara, CA, USA), as described above, with the same mutagenesis primers. WT, Y67F, and Y67E Cyt*c* expression constructs as well as a negative control empty vector (EV) construct were stably transfected into Cyt*c* double-knockout mouse lung fibroblasts [70] (a kind gift from Dr. Carlos Moraes, University of Miami, Coral Gables, FL, USA) using Transfast transfection reagent (Promega; Madison, WI, USA) in a 1:1 transfection reagent-to-DNA ratio, according to the manufacturer’s protocol. The transfected cells were cultured in DMEM supplemented with 10% FBS (Sigma-Aldrich; St. Louis, MO, USA), 1% Pen/Strep, 1 mM sodium pyruvate, and 50 µg/mL uridine at 37 °C in 5% CO_2_. Positive clones were selected in the presence of 2 μg/mL puromycin. After selection, the clones were cultured in DMEM supplemented with 10% FBS and 1% Pen/Strep for all experiments unless otherwise stated.

### 2.14. Western Blotting for Cytc

Cyt*c* expression levels of intact cells expressing the Cyt*c* variants were assessed via western blot after SDS-PAGE following our standard protocol [58]. After protein transfer to a PVDF membrane (#1620177, Bio-Rad; Hercules, CA, USA) and blocking with 5% non-fat dry milk in 1x TBS-T and 0.1% Tween 20 for 1 h at room temperature, the membrane was incubated with a 1:4000 dilution of mouse anti-Cyt*c* antibody (#556433, BD Pharmingen; San Jose, CA, USA) in a blocking reagent overnight at 4 °C. A second gel was run and processed identically to the above, with the second membrane being incubated with a 1:5000 dilution of mouse anti-GAPDH antibody (#60004-1-Ig, Proteintech; Rosemont, IL, USA) as a loading control in blocking solution overnight at 4 °C. The next day, the membranes were incubated with a 1:5000 or 1:10,000 dilution of sheep anti-mouse IgG conjugated to horseradish peroxidase secondary antibody (#NA931V, GE Healthcare) in blocking reagent for 2 h at room temperature. The blots were visualized using Pierce ECL western blot substrate (#32106, Thermo Fisher Scientific).

### 2.15. Measurement of Oxygen Consumption Rate in Intact Cells

The oxygen consumption rates (OCRs) of the intact cells expressing the Cyt*c* variants were measured. Cells were seeded at 15,000 cells/well in a Seahorse XF^e^24 cell culture microplate and cultured overnight in 250 μL/well growth media. After 20 h, the growth media was replaced with Seahorse XF media, pH 7.4, supplemented with 10 mM glucose and 1 mM sodium pyruvate. After the media change, cells were incubated in a CO_2_-free incubator for 1 h, then intact cellular respiration was measured in an XF^e^24 Seahorse extracellular flux analyzer (Seahorse Bioscience; North Billerica, MA, USA). After the experiment, the cells in each well were lysed with 25 μL RIPA buffer containing a protease inhibitor cocktail (Roche Applied Science; Mannheim, Germany) and the protein concentration was determined using the DC protein assay (Bio-Rad; Hercules, CA, USA) which was standardized to a bovine serum albumin standard curve. OCR is reported as pmol O_2_/minute/μg protein.

### 2.16. Measurement of Mitochondrial Membrane Potential

The relative mitochondrial membrane potentials (ΔΨ_m_) of the intact cells expressing the Cyt*c* variants were measured. The cells were seeded at 15,000 cells/well onto black 96-well plates (Costar, CLS3603; Sigma, Ronkonkoma, NY, USA) and cultured overnight. To assess for relative changes in ΔΨ_m_, the cells were incubated for 1 h at 37 °C with 1 μM JC-10 (Molecular Probes, Inc., Eugene, OR, USA) in DMEM without phenol red or FBS. Green (excitation/emission of 485 nm/527 nm) and red (excitation/emission of 485 nm/590 nm) fluorescence was measured in PBS using a Synergy H1 microplate reader (BioTek Instruments Inc.; Winooski, VT, USA). ΔΨ_m_ is reported as the ratio of red–green fluorescence.

### 2.17. Measurement of Mitochondrial ROS

The production of mitochondrial ROS of the intact cells expressing the Cyt*c* variants was measured. Cells were seeded onto 24-well plates and cultured overnight. To assess for changes in mitochondrial ROS production, the cells were incubated for 20 min at 37 °C with 5 μM MitoSOX (M36008, Thermo Scientific) in DMEM without phenol red or FBS. Fluorescence was measured in PBS using a Synergy H1 microplate reader (BioTek Instruments Inc.) with excitation/emission wavelengths of 510 nm/580 nm. After the experiment, the cells in each well were lysed with RIPA buffer containing a protease inhibitor cocktail (Roche Applied Science; Mannheim, Germany) and the protein concentration was determined using the DC protein assay (Bio-Rad; Hercules, CA, USA) which was standardized to a bovine serum albumin standard curve. Mitochondrial ROS production is reported as a percentage of change in the fluorescence/μg protein (arbitrary units/μg protein) compared to WT.

### 2.18. Cell Death Using Annexin V/Propidium Iodide Staining

The cell death levels of the intact cells expressing the Cyt*c* variants were measured following exposure to H_2_O_2_ or staurosporine using annexin V/propidium iodide (PI). Cells (1 × 10^6^) were seeded on 10 cm cell culture dishes and cultured overnight in growth media. The next day, the cells were exposed to either H_2_O_2_ (300 μM for 14 h) or staurosporine (1 μM for 5 h). After treatment, the cells were detached using 1 mM EDTA in PBS at 37 °C for 2 min. The cells were washed twice with cold PBS and resuspended in 1× binding buffer from the FITC annexin V apoptosis detection kit I (BD Pharmingen). A total of 300 μL of cell suspension was incubated for 15 min with 7 μL annexin V and 5 μL PI in a light-protected environment. Binding buffer (2 mL) was added to each tube. A CyFlow Space flow cytometer (Sysmex America, Inc.; Lincolnshire, IL, USA) was used to record fluorescence signals. The data were analyzed with FCS Express 7 software (De Novo Software; Glendale, CA, USA). Cell death is reported as a percentage of cells.

### 2.19. Statistical Analyses

Data are presented as means, and error bars indicate standard deviation. Statistical analyses of the data were performed using GraphPad Prism v10.4.0 (GraphPad Software; San Diego, CA, USA). The data were analyzed using one-way ANOVA comparing the mean of each column with the mean of the control column (WT) using the Dunnett post hoc test. Statistics for the COX activity assay were performed as described above specifically on the 15 μM Cyt*c* condition. Statistics for the Heme degradation assay were performed as described above, specifically at the 800 s timepoint. Statistics for the Annexin V/PI assays were performed as described above, specifically for total cell death (PI+ cells, annexin V+ cells, annexin V+/PI+ cells combined). *p* values are indicated in the figures.

## 3. Results

### 3.1. Cytc Y67 Is Conserved in Mammals and Phosphorylated in Bovine Heart

Continuing our ongoing investigation of Cyt*c* phosphorylation [47,50,52,57], we purified Cyt*c* from snap-frozen bovine heart tissue. Mass spectrometric analysis unambiguously revealed a novel phosphorylation site on Y67 of Cyt*c* (Figure 1A). The side chain of Y67 is pointed toward the heme cavity (Figure 1B), suggesting that phosphorylation at this site may impact heme-related functions of Cyt*c*. Residue Y67 is highly conserved in mammals and other clades [77], further suggesting a functional relevance in Cyt*c* regulatory roles for this phosphorylation (Figure 1C).

### 3.2. Overexpression and Purification of Functional Cytc Variants in E. coli Cells

Rat and mouse Cyt*c* share identical amino acid sequences. Cyt*c* wild-type (WT) and mutant expression plasmids were constructed based on the rat cDNA sequence, and Y67 was replaced with phosphomimetic glutamate (Y67E) or the non-phosphorylatable control phenylalanine (Y67F). Although glutamate is shorter, less bulky, and lacks aromaticity compared to phosphorylated tyrosine (Figure 2A), using a glutamate replacement as a phosphomimetic to study phosphorylation is an established strategy for the characterization of both serine and tyrosine phosphorylation, including specifically for Cyt*c* [47,49,50,51,53,54] and other proteins [78]. The advantage of using recombinant phosphomimetic Y67E Cyt*c* is that it can be easily purified in large amounts after overexpression in bacteria, it maintains the negatively charged side chain unlike phospho-tyrosine which can easily be lost during experiments, and it simulates the full occupancy of the phosphorylation. The expression plasmids for WT, Y67F, and phosphomimetic Y67E Cytc were overexpressed in *E*. *coli* C41 (DE3) cells, and the Cyt*c* variants were purified. The Coomassie blue-stained gel shows that Cyt*c* variants have been purified to homogeneity (Figure 2B).

### 3.3. Cytochrome c Oxidase Activity Is Inhibited in the Reaction with Phosphomimetic Y67E Cytc

The first major function of Cyt*c* is as an electron carrier between complex III and COX in the ETC. Our previous studies have shown decreased activity of other phosphorylation sites of Cyt*c* in the reaction with COX [47,49,50,52,79]. To examine if phosphomimetic Y67E Cyt*c* has a similar effect, we analyzed the activities of the Cyt*c* variants in the reaction with purified pig heart COX, which was isolated under conditions preserving the in vivo regulatory properties of the enzyme [69]. COX activity in reaction with 15 µM phosphomimetic Y67E Cyt*c* was 21% reduced compared to the WT (Figure 2C). The COX activity values at 15 µM were 13.28 s^−1^ for the WT, 15.16 s^−1^ for Y67F, and 10.43 s^−1^ for phosphomimetic Y67E Cyt*c*. Interestingly, the K_m_ value was significantly increased for the phosphomimetic Y67E Cyt*c*. The K_m_ values were 1.1 μM for the WT, 0.6 μM for Y67F, and 3.4 μM for phosphomimetic Y67E Cyt*c*.

### 3.4. Caspase-3 Activity Is Inhibited by Phosphomimetic Y67E Cytc

The second major function of Cyt*c* is to initiate intrinsic apoptosis, when it is released from the mitochondria. Cyt*c* then binds to Apaf-1 in the cytosol, which activates caspase-9 leading to the activation of caspase-3. To test the effect of phosphomimetic Y67E Cyt*c* on apoptosis, we analyzed downstream caspase-3 activity using a cell-free caspase-3 activity assay. Phosphomimetic Y67E Cyt*c* showed 51% decreased caspase-3 activity compared to the WT (Figure 2D).

### 3.5. Redox Potential, Susceptibility to Oxidation, and Superoxide Scavenging Activity Are Decreased in Phosphomimetic Cytc

Cyt*c* redox potential values are reported to be in the range of 220–270 mV [71,80], allowing for effective electron transfer between complex III and COX. The enzyme midpoint redox potential of purified Cyt*c* variants was measured spectrophotometrically via the equilibration method using 2,6-dichloroindophenol (DCIP) as a reference compound. The measured values were 235 mV for the WT, 208 for Y67F, and 166 mV for phosphomimetic Y67E Cyt*c* (Figure 2E). The phosphomimetic Y67E was 29% decreased compared to the WT, indicating that Y67 is important for the redox properties of the heme moiety.

To determine the rate of Cyt*c* oxidation, the Cyt*c* variants were fully reduced with sodium dithionite, followed by the removal of the reductant. The ferro-Cyt*c* variants were oxidized in the presence of H_2_O_2_. The phosphomimetic Y67E Cyt*c* demonstrated 23% decreased rate of oxidation compared to the WT (Figure 3A).

To determine the rate of Cyt*c* reduction by superoxide, the Cyt*c* variants were fully oxidized with potassium ferricyanide, followed by the removal of the oxidant via NAP-5 columns. The ferri-Cyt*c* variants were reduced in the presence of superoxide generated by the hypoxanthine–xanthine oxidase system. Superoxide dismutase (SOD) was used as a negative control by neutralizing superoxide. The reduction rate of phosphomimetic Y67E Cyt*c* was 90% decreased compared to the WT (Figure 3B), suggesting that phosphomimetic replacement of Y67 strongly inhibits the superoxide scavenging ability of the Cyt*c*.

### 3.6. Phosphomimetic Y67E Cytc Showed Increased Susceptibility to Degradation Under High Levels of Oxidative Stress

High levels of H_2_O_2_ are able to degrade the heme group of Cyt*c*, which may be impacted by Y67 phosphorylation due to its proximity to the heme group. The degradation of Cyt*c* variants was tested via exposure to high concentrations of H_2_O_2_. At 800 s after exposure, only 15% of the phosphomimetic Y67E Cyt*c* was still intact compared to 54% of the WT (Figure 3C), suggesting that phosphomimetic replacement on Y67 destabilizes the heme group.

### 3.7. Phosphomimetic Y67E Cytc Impairs Cardiolipin Peroxidase Activity

Cyt*c* peroxidation of cardiolipin (CL) is a secondary pro-apoptotic function of Cyt*c*. Therefore, we examined the peroxidase activity of the Cyt*c* variants. In the presence of H_2_O_2_, Cyt*c* catalyzes the peroxidation of CL, which then oxidizes Amplex Red to form the fluorescent product, resorufin. The phosphomimetic Y67E Cyt*c* displayed a minimal ability to catalyze the peroxidation of CL (Figure 3D, indicating that Y67 may be crucial for this function of the Cyt*c*.

### 3.8. Molecular Dynamics Modeling of Cytc

To explore possible differences in the Cyt*c* Y67 variants in their solution structures, molecular dynamics simulations were performed using the A chain from WT Cytc (PDB entry 5C0Z). For phosphorylated Y67, Y67F, and Y67E Cyt*c* variants, the Y67 position of the WT Cytc structure was replaced with the specified modification or mutation. In all simulations of the amino acid side chains, the final solution structures of Cyt*c* after 600 and 700 ns remained stable. Interestingly, the loop consisting of amino acids 20–30, a region known to be flexible, showed lower root mean square fluctuations (RMSFs) for phosphorylated Y67 and phosphomimetic Y67E Cyt*c* compared to the WT (Figure 4), suggesting that the protein loses some of its flexibility.

### 3.9. Cells Expressing Phosphomimetic Y67E Cytc Have Decreased Mitochondrial Respiration

To test the effect of phosphomimetic substitution of Y67 on mitochondrial respiration in intact cells, Cyt*c* double-knockout mouse embryonic fibroblasts, where both the somatic and testes isoforms are knocked out, were used to generate stable cell lines expressing the Cyt*c* variants. Cyt*c* expression constructs were transfected into the Cyt*c* double-knockout cells. Stable cell line clones with equivalent expression levels of each Cyt*c* variant were selected for further analysis (Figure 5A). As an additional control, the cells were also transfected with an empty vector (EV) construct which lacks the sequence coding for Cyt*c*. The basal oxygen consumption rate (OCR) was measured using a Seahorse bioanalyzer. The cell line expressing phosphomimetic Y67E Cyt*c* displayed a 35% reduction in respiration rates compared to the WT (Figure 5B).

### 3.10. Cells Expressing Phosphomimetic Y67E Cytc Have Decreased Mitochondrial Membrane Potential and ROS Production

Due to a decreased mitochondrial respiration rate observed in cells expressing phosphomimetic Y67E Cyt*c*, we next examined ΔΨ_m_ and ROS generation in cells expressing the Cyt*c* variants, given the known relationship between respiration, ΔΨ_m_, and ROS generation. Relative ΔΨ_m_ levels were measured using JC-10, a voltage-dependent probe which is represented by the ratio of red to green fluorescence indicative of high and low ΔΨ_m_, respectively. In cells expressing phosphomimetic Y67E, ΔΨ_m_ was decreased as indicated by a 39% reduction in red to green fluorescence, compared to the WT (Figure 5C), which matches the pattern of the corresponding respiration rates. As high levels of ΔΨ_m_ lead to mitochondrial ROS production, we examined ROS levels using MitoSOX, a mitochondrial ROS indicator. As expected, ROS levels were reduced in cells expressing the phosphomimetic Y67E Cyt*c*, as indicated by a 39% reduced fluoresce signal compared to WT (Figure 5D).

### 3.11. Cells Expressing Phosphomimetic Y67E Cytc Show Decreased Apoptosis After Cellular Injury

Our in vitro data showed that phosphomimetic Y67E Cyt*c* decreased the activation of caspase-3 compared to the WT in the cell-free detection system. To assess apoptosis in the cell culture model, we analyzed apoptosis via annexin V/propidium iodide staining and flow cytometry of the cells expressing the Cyt*c* variants. Apoptosis was initiated with exposure to H_2_O_2_ or staurosporine. After treatment with H_2_O_2_, cells expressing phosphomimetic Y67E Cyt*c* demonstrated 15% of total cell death compared to 27% of total cell death for the WT (Figure 6A,C). Similarly, after treatment with staurosporine, cells expressing phosphomimetic Y67E Cyt*c* demonstrated 19% of total cell death compared to 31% of total cell death for the WT (Figure 6B,D). Additionally, cells transfected with empty vector displayed the lowest rate of cell death in both experiments compared to the other Cyt*c* variants due to the absence of Cyt*c*.

## 4. Discussion

Cyt*c* is a central player in mitochondrial metabolism and intrinsic apoptosis. We have proposed that the reaction between Cyt*c* and COX is the rate-limiting step of the ETC and is regulated by post-translational modifications of Cyt*c* [79]. Although conservation is not necessarily indicative of functional relevance, Y67 is one of the 15 amino acid residues that are highly conserved and contribute significantly to the structure, function, folding, and stability of Cyt*c* [77,81,82]. As one of the better-studied residues in Cyt*c*, Y67 is known to allow Cyt*c* to switch between native and alternative conformations to meet its role in multiple functions [83]. A wide body of research suggests that Y67 plays a crucial role in the hydrogen bonding network of N52, T78, and M80, which is eventually relayed to the heme group (reviewed in [77]). Disruption to this hydrogen bonding network has previously been shown to interfere with the functional properties of Cyt*c* [83,84,85,86,87,88]. Y67 has also been identified as a key residue for catalyzing CL peroxidation via the formation of a tyrosyl radical as an intermediate during the oxidation reaction [81], as we confirmed here. Furthermore, Y67 nitration has previously been shown to reduce state 4 respiration in rat heart mitochondria and rate of reduction with ascorbate [89] as well as impaired caspase activation [90]. Mutagenesis studies targeting Y67 have found that Y67H and Y67R increase peroxidase activity [84,85]. Interestingly, mutagenesis studies on Y67F have found that the mutation protects the heme group from degradation, allows a second water molecule into the pocket near the heme group, and leads to reduced midpoint redox potential [86,88]. Additionally, Y67 phosphorylation has also been reported by high-throughput phosphoproteomics studies looking at several lung cancer cell lines [91]. However, the role of Y67 phosphorylation has never previously been characterized.

In the present study, we provide evidence that Y67 phosphorylation of Cyt*c* plays a pivotal role in its function as a regulator in respiration, apoptosis, ROS scavenging, and redox stability (Figure 7). Previously, we reported that Y97 and Y48 are phosphorylated in Cyt*c* purified from cow heart and liver in vivo, respectively, which led to inhibition of respiration in the reaction with COX [52,57]. Additionally, Y48 phosphorylation abolished the function of Cyt*c* to trigger caspase 3 activation [53]. These previous works, along with the data presented here, indicate that tyrosine phosphorylation is an important regulator of Cyt*c* function. Generally, tissue-specific phosphorylations of Cyt*c* tend to be present under basal conditions, inhibiting respiration and apoptosis (reviewed in [23]). These phosphorylations provide metabolic adaptations to maintain optimal tissue functioning while also making the tissues more resilient to physiological stressors and cell death. The loss of these phosphorylations during ischemia, likely due to a loss of mitochondrial calcium homeostasis and activation of phosphatases, results in ΔΨ_m_ hyperpolarization during reperfusion. This then drives increased ETC activity, allowing reverse electron transfer in the ETC and ROS bursts (reviewed in [79]). We have previously proposed that the loss of these inhibitory phosphorylations during ischemia is an ineffectual effort by the cell to upregulate ETC activity to increase ATP levels; however, at the same time sensitizing the tissue to apoptosis as an unintended consequence when oxygen reenters the tissue during perfusion [58]. Y67 phosphorylation of Cyt*c*, which we report here for bovine heart tissue under basal conditions, may contribute to ischemia–reperfusion injury of the heart during myocardial infarction similar to that of other phosphorylations of Cyt*c*, such as S47 phosphorylation in brain tissue [50,51]. A limitation of our study is that heart Cyt*c* is also phosphorylated on Y97. Thus, using purified Cyt*c* from heart for in vitro studies of Y67 phosphorylation would be overshadowed by the presence of Y97 phosphorylation. As such, focusing on just a single site, we characterized the impact of Y67 phosphorylation on the various functions of Cyt*c* indirectly through phosphomimetic replacement.

Like our previous data studying phosphorylation of Cyt*c*, phosphomimetic Y67E Cyt*c* inhibits oxygen consumption in the reaction with COX, which may be partially explained by the perturbation of the hydrogen bond between Y67 and M80, as well as decreases in the redox potential. Intact cells expressing phosphomimetic Y67 Cyt*c* also demonstrated reduced respiration rates as analyzed by the Seahorse assay with corresponding reductions in ∆Ѱ_m_ and mitochondrial ROS. Interestingly, phosphomimetic Y67E Cyt*c* also showed reduced rates of reduction and oxidation as well as strongly decreased resistance to heme degradation, suggesting that Y67 phosphorylation increases Cyt*c* susceptibility to redox stress.

Phosphorylation of Cyt*c* has previously been shown to inhibit the pro-apoptotic functions of Cyt*c* on some of the other residues that can be phosphorylated [5,22,53]. This has partially been attributed to the phosphorylation causing reduced electrostatic interactions between the positively charged Cyt*c* to its negatively charged binding pocket on Apaf-1 [92,93]. Such a decrease was also observed with the phosphomimetic Y67E Cyt*c*, which showed decreased downstream caspase-3 activity. Similarly, cells expressing phosphomimetic Y67E Cyt*c* showed decreased levels of total cell death after treatment with H_2_O_2_ or staurosporine. Another pro-apoptotic function of Cyt*c* is catalyzing the peroxidation of CL [16]. Y67 of Cyt*c* is one of the four conserved tyrosine residues present in close proximity to the heme group and is the likely tyrosine radical site for peroxidation of CL [77,94]. Phosphomimetic replacement nearly completely abolished the ability of Cyt*c* to catalyze the peroxidation of CL, indicating that this residue plays a central role for this function. The molecular dynamics simulations also offer a possible explanation for this functional change. Previously, we reported the T28 loop epitope as displaying the highest flexibility within Cyt*c* [47,49,50,61]. However, in this publication, we see less mobility within the T28 loop of phosphomimetic Y67E Cyt*c*. Typically, the omega loops of Cyt*c* show increased dynamics during CL binding and peroxidation [95]. The reduced mobility suggested for this region in phosphomimetic Cyt*c* may impair CL binding affinity and CL peroxidase activity. Additionally, cells expressing phosphomimetic Y67E Cyt*c* showed decreased levels of total cell death after exposure to H_2_O_2_ or Staurosporine. Overall, these data suggest that Y67 and its modification, given its proximity to the heme group, play an important role in modulating the metabolic and apoptotic properties of Cyt*c*.

In our studies, we also included the Y67F control variant, which cannot be phosphorylated. Phenylalanine only differs by the absence of the hydroxyl group from tyrosine. Because the hydroxyl group is part of the hydrogen bonding network connecting Y67 to the heme group as discussed above, it is not surprising that its absence caused some functional changes. In most experiments Y67F Cyt*c* behaved in a way between the WT and Y67E Cyt*c*, such as caspase-3 activation, redox potential, and cardiolipin peroxidation, which is mirrored by the molecular dynamics simulations that produced a pattern in between the WT and Y67E Cyt*c*. Although not statistically significantly different from the WT, in the reaction with COX, Y67F Cyt*c* showed a trend toward higher activity, suggesting that the presence of the hydroxyl group has a mild inhibitory effect, whereas the Y67E replacement produced a more pronounced significant inhibitory effect.

In conclusion, we show that Cyt*c* from heart can be phosphorylated on Y67 in addition to the previously reported phosphorylation of Y97. Functional characterization by phosphomimetic Y67E replacement revealed mostly inhibitory effects on the various functions of Cyt*c*, with reductions seen in COX activity, downstream caspase-3 activity, cardiolipin peroxidase activity, and redox potential. Cells expressing phosphomimetic Cyt*c* show lowered respiration, ∆Ѱ_m_, ROS production, and apoptosis. We propose that these changes are beneficial and tissue-protective by regulating mitochondrial respiration to maintain optimal intermediate ∆Ѱ_m_ levels for effective ATP production and, simultaneously, preventing ∆Ѱ_m_ hyperpolarization that would lead to excessive ROS generation. Future research should identify the underlying signaling pathway including immediately upstream kinases and phosphatases that (de-)phosphorylate Cyt*c*. Targeting those signaling molecules by inhibiting the phosphatase and/or activating the kinase could be therapeutically useful under conditions of stress such as ischemia–reperfusion (e.g., myocardial infarction), when these protective modifications are lost, causing excessive ROS damage and cell death.

## Figures and Tables

**Figure 1 cells-14-00951-f001:**
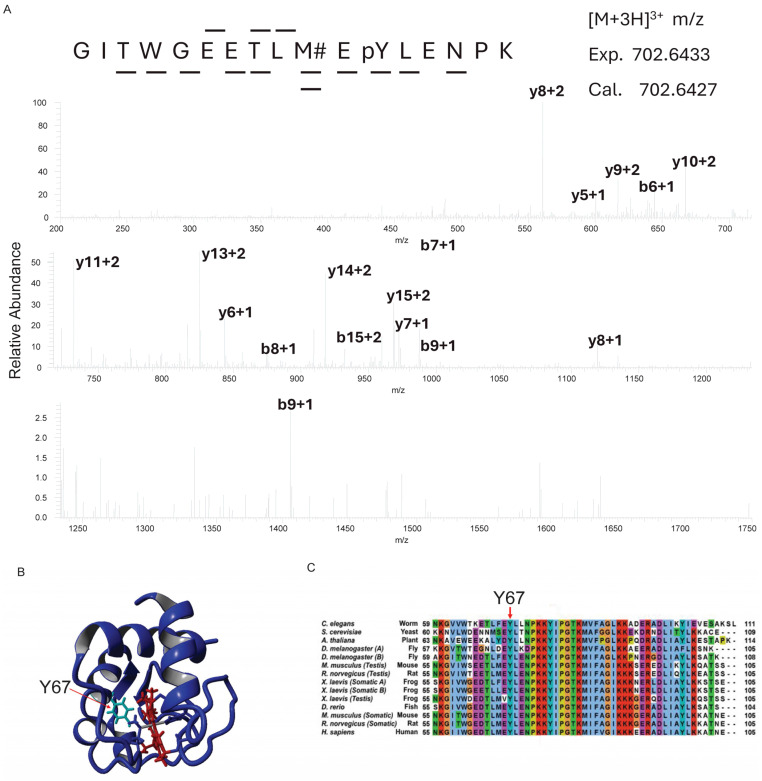
(**A**) Representative mass spectrum of the bovine heart Cyt*c* peptide GITWGEETLMEY (phospho)LENPK identifying Cyt*c* Y67 phosphorylation (*n* = 3). Y67 phosphorylation was assigned by fragment ions y^5^, y^6^, y^7^, y^8^, y^10^, y^11^, y^13^, y^14^, and y^15^. The peptide sequence was identified by fragment ions y^3^, y^5^, y^6^, y^7^, y^8^, y^10^, y^11^, y^13^, y^14^, y^15^, b^6^, b^8^, and b^9^. The Mascot MOWSE score, which is a probability-based score of the statistical significance for a peptide match versus a random occurrence, was 69.6 for this peptide, providing strong support for the assignment. (**B**) Crystal structure (5C0Z.pdb) of rodent WT Cytc (blue) with the heme group (red) and Y67 (cyan) labeled. (**C**) Alignment of Cyt*c* with eukaryotes using Clustal X.

**Figure 2 cells-14-00951-f002:**
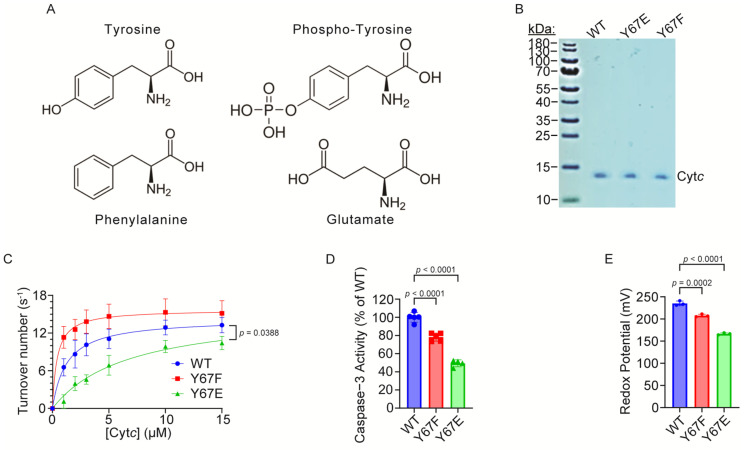
(**A**) Chemical structure diagrams of tyrosine, phospho-tyrosine, phenylalanine, and glutamate showing the structural differences between the residues. (**B**) Coomassie blue staining of the recombinant Cyt*c* variants on a 10% tris-tricine SDS-PAGE gel showing purity. (**C**) Oxygen consumption rate of 26.7 nM cow heart COX measured using an Oxygraph system at 25 °C in reaction with the recombinant Cyt*c* variants (*n* = 3–4). (**D**) Downstream caspase-3 activity of recombinant Cyt*c* variants (*n* = 4–5). (**E**) Redox potential of recombinant Cyt*c* variants (n = 3).

**Figure 3 cells-14-00951-f003:**
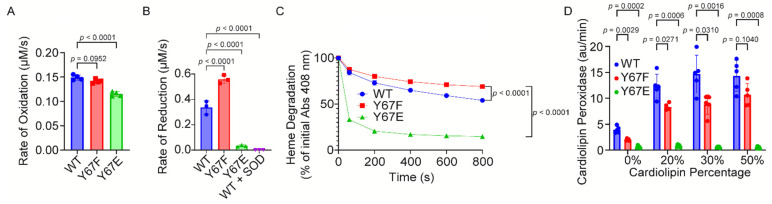
(**A**) Initial rate of the oxidation of reduced recombinant Cyt*c* variants by 100 μM H_2_O_2_ (*n* = 4). (**B**) Superoxide scavenging of oxidized Cyt*c*. Shown are the initial rates of the reduction of oxidized recombinant Cyt*c* variants by superoxide generated by a hypoxanthine/xanthine oxidase system (*n* = 3). (**C**) Heme degradation of recombinant Cyt*c* variants by 3 mM H_2_O_2_. (**D**) Cardiolipin peroxidase activities of recombinant Cyt*c* variants with lipid mixtures containing 0%, 20%, 30%, and 50% cardiolipin were measured after the addition of 5 μM H_2_O_2_ (*n* = 5). Data are represented as mean ± standard deviation.

**Figure 4 cells-14-00951-f004:**
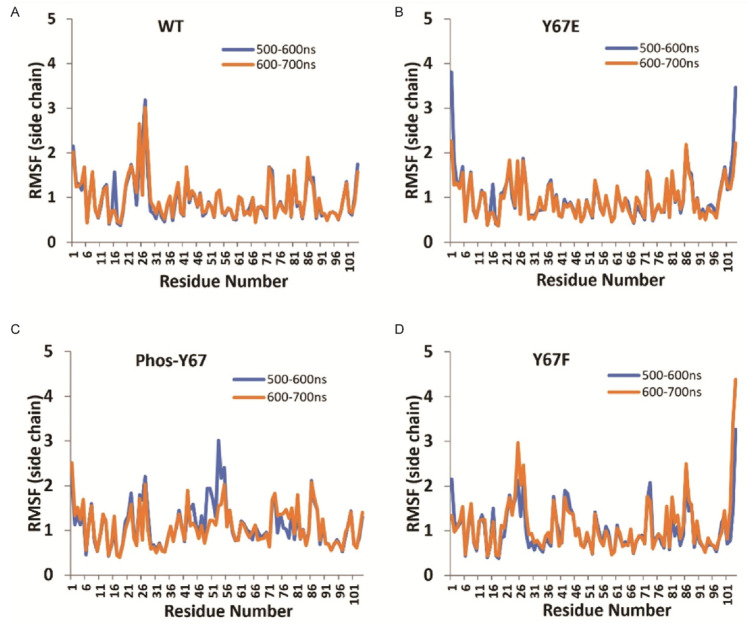
(**A**) Root mean square fluctuations (RMSFs) of amino acid side chains in Å of WT Cytc were calculated for 500 to 600 ns (blue) and 600 to 700 ns (orange) intervals using Molecule A from the WT Cytc crystal structure. (**B**) Equivalent to Figure 4A for Y67F Cyt*c*. (**C**) Equivalent to Figure 4A for phospho-Y67 Cyt*c*. (**D**) Equivalent to Figure 4A for Y67E Cyt*c*.

**Figure 5 cells-14-00951-f005:**
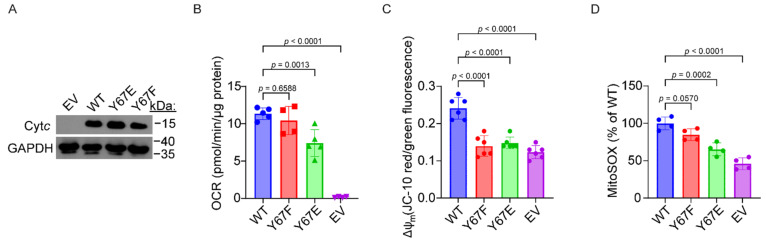
(**A**) Representative western blot of Cyt*c* double-knockout cells transfected with Cyt*c* variants which shows equivalent Cyt*c* expression with GAPDH as a loading control. (**B**) Basal oxygen consumption rate (OCR) of cells expressing Cyt*c* variants (*n* = 4–5). (**C**) Mitochondrial membrane potential (ΔΨ_m_) of cells expressing Cyt*c* variants (*n* = 6). (**D**) Mitochondrial ROS production of cells expressing Cyt*c* variants (*n* = 4). The data are represented as mean ± standard deviation.

**Figure 6 cells-14-00951-f006:**
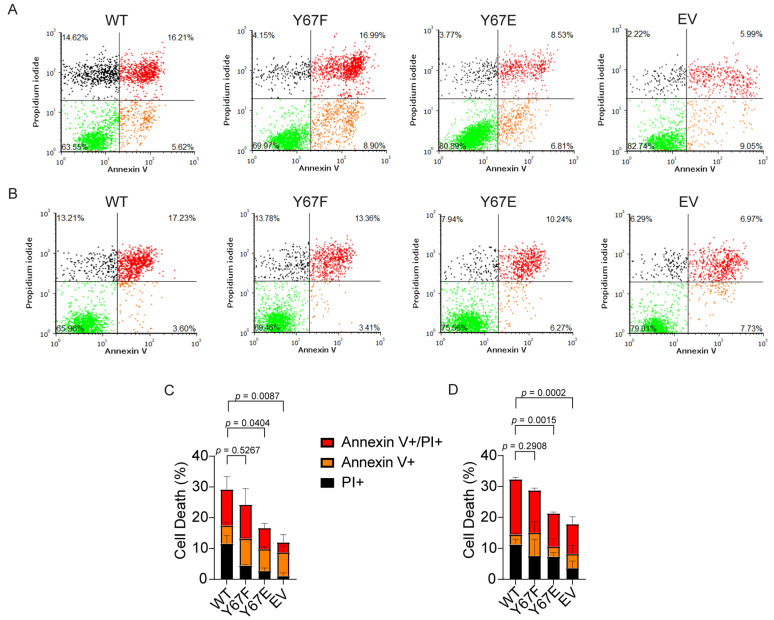
(**A**,**C**) Annexin V/propidium iodide flow cytometric analyses after exposure to 300 μM H_2_O_2_ for 14 h (*n* = 3). (**B**,**D**) Annexin V/propidium iodide (PI) flow cytometry data after exposure to 1 µM staurosporine for 5 h (*n* = 3). Data are represented as means ± standard deviation.

**Figure 7 cells-14-00951-f007:**
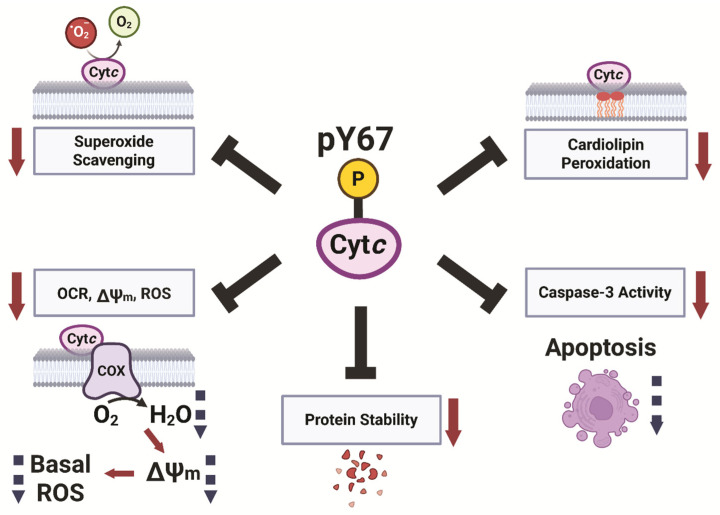
Model of Cytc regulation through Y67 phosphorylation.

## Data Availability

The original contributions presented in this study are included in the article. Further inquiries can be directed to the corresponding author.

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
