# Peer review of "Tyrosine 67 Phosphorylation Controls Respiration and Limits the Apoptotic Functions of Cytochrome c"

_cells, 2025, doi:10.3390/cells14130951_

Round 1
Reviewer 1 Report
Comments and Suggestions for Authors
This manuscript describes the role of Tyr 67 phosphorylation of cytochrome c in cell functioning including respiratory chain activity and apoptosis. The manuscript has complete and detailed information on cytochrome c mutants where Tyr 67 is mutated to Glu (to mimic phosphorylation) and also Phe (to remove charge and have some steric similarity). The Glu mutant cytochrome c showed inhibition of cytochrome oxidase activity and also inhibited pro-apoptotic activities. In intact knockout cells, the Glu mutant showed lowered cell respiration with accompanying reductions in membrane potential, and reactive oxygen species.
The following changes should be included in a revised manuscript:
- The method while explicit and well written should be trimmed to 1/3 of its current length.
As many of the methods are already published, the length of this section is unnecessary;
- There needs to be short explanation of the size relationship between phosphorylated Tyr and replacement mutants Glu and Phe in a three-dimensional structure;
- In Figure 2, why are the turnover numbers for cytochrome oxidase so low? Does the assay buffer contain Tween 20? A turnover number 15 sec-1 at pH 7.4 extremely low as it should be 170-200 sec-1;
- This is a semantic point. It is unlikely that 50% cardiolipin/phospholipid mixture are liposomes. Perhaps a better term would be a cardiolipin/phospholipid mixture;
- The molecular dynamics section can be stated in one sentence and the figure 4 can be omitted.
- In Figure 5C, the Delta psi in the mutants is the same as the empty vector yet SOX activity is increased in the mutants over the empty vector. Explanation?;
- A reference for mascot moss score and what it means should be included in the manuscript.
- Typos In figure 5 legend the letter C is missing. On Page 16, below fig 7 the line beginning with “plained …..hydrogen bond not bound”
In summary, this manuscript has complete information about the role of cytochrome c Tyr 67 and its phosphorylation in cell functioning using site-directed mutagenesis. This large amount of work should be published.
Author Response
- The method while explicit and well written should be trimmed to 1/3 of its current length. As many of the methods are already published, the length of this section is unnecessary.
Response: Another reviewer commented asking for more specificity in the methods section (see reviewer 2 comment 3). We have tried to balance that feedback with the feedback given here to decrease the length of the methods section. This has resulted in significant cuts to the longer methods subsections, specifically “2.1.”, “2.3.”, “2.6.”, “2.16.” and “2.18.”.
- There needs to be a short explanation of the size relationship between phosphorylated Tyr and replacement mutants Glu and Phe in a three-dimensional structure.
Response: Newly added Figure 2A shows the structures of the amino acids in comparison to phosphorylated tyrosine. We added a sentence in section 3.2. acknowledging the glutamate is slightly shorter, less bulky, and not aromatic in comparison to phosphorylated tyrosine.
- In Figure 2, why are the turnover numbers for cytochrome oxidase so low? Does the assay buffer contain Tween 20? A turnover number 15 sec-1 at pH 7.4 is extremely low as it should be 170-200 sec-1.
Response: There are several contributing factors as to why the COX assay activity is low. First, we are using rodent Cytc with porcine COX. Mismatch between the species likely contributes to reduced activity. Second, the protocol we use for COX purification, unlike other groups, retains a larger proportion of the post-translational modification to the complex, which have previously been shown to be inhibitory. Third, this was old enzyme previously purified approximately a decade ago. The assay buffer does contain 1% Tween-20. While Tween-20 is generally considered to be a gentle surfactant, it is possible that there is also reduction in COX activity due to this.
- This is a semantic point. It is unlikely that 50% cardiolipin/phospholipid mixture are liposomes. Perhaps a better term would be a cardiolipin/phospholipid mixture.
Response: We have changed “liposome” to “cardiolipin/phospholipid mixture” or “lipid mixture”.
- The molecular dynamics section can be stated in one sentence and the figure 4 can be omitted.
Response: Although the molecular dynamics simulations do not suggest a dramatic change, we believe that these results are worth presenting because they support the use of the phosphomimetic Y67E replacement since Y67E and phospho-Y67 at 600-700 ns show no dramatic differences. Therefore, we would prefer to keep this figure.
- In Figure 5C, the Delta psi in the mutants is the same as the empty vector yet SOX activity is increased in the mutants over the empty vector. Explanation?;
Response: While mitochondrial membrane potential and mitochondrial ROS production are typically proportional, the relationship is still complex. The mutants have an intact, but inhibited, electron transport chain, while the empty vector does not have an intact electron transport chain. Mitochondrial membrane potential is a crucial intracellular signal, so we hypothesize that, despite the dysfunction of the overall electron transport chain in empty vector mitochondria, certain complexes within empty vector mitochondria run in reverse in order to generate some mitochondrial membrane potential. Alternate pathways, such as this, contribute to generating a higher mitochondrial membrane potential than would be expected for a completely non-functional electron transport chain pathway.
- A reference for mascot moss score and what it means should be included in the manuscript.
Response: The Mascot MOWSE score is a statistical value assigned to a peptide indicating how real the data a versus a random occurrence. The score reported here is excellent. We have explained the meaning of the score in the figure legend where the score is mentioned.
- Typos: In figure 5 legend the letter C is missing. On Page 16, below fig 7 the line beginning with “plained …..hydrogen bond not bound”
Response: We appreciate catching the typos and have corrected them.
Reviewer 2 Report
Comments and Suggestions for Authors
The work represents a significant contribution in the field of functional characterisation of phosphorylation at position Y67 in vivo, with additional explanation of the metabolic consequences in combination with in vitro cell models. The value of the work lies in the fact that it successfully maps the in vivo phosphorylation of Y67 and opens up new perspectives for the understanding of the associated metabolic processes. Thus, the work offers new insights into this topic, which is its greatest scientific strength.
However, despite these contributions, the paper has a number of structural and substantive weaknesses that detract from its clarity and scientific basis. The main shortcoming of the paper is the lack of a clearly defined purpose and aim of the research. The authors do not mention specific hypotheses, or the potential translational or applied value of the results obtained. This makes it difficult for the reader to understand either the guiding principles of the research or the scientific motivation behind the work.
The introductory part of the text is generally well written but needs some refinement. Firstly, the purpose and aim of the research are not clearly formulated. In addition, the last paragraph of the introduction contains data closer to the presentation of the results and their interpretation, which is inappropriate for the introduction from a methodological point of view; this information would be better included in the results and/or discussion section.
The methodological section contains several omissions that affect the transparency and reproducibility of the experiment. For example, the colouring of the gel with Coomassie blue is not stated, although it can be inferred from the context that it was used. Important technical data is also missing, such as the number of heart samples analysed, the manufacturers of the ion exchangers (DE52 – anionic and CM52 – cationic) and the manufacturer of the spectrophotometer used (Jasco V-570). This information is of crucial importance for the transparency and reproducibility of the analysis and should therefore be included.
The results section is detailed and informative but contains technical and content-related irregularities. The principle of certain methods is described several times unnecessarily, which should be omitted in order not to overload the text. Some descriptions of the results are repeated, e.g. ˝The next day the cells were exposed to either H2O2 (300 μM for 14 h) or staurosporine (1 μM for 5 h) ˝. Figure 2D is not mentioned in the text, although it is supposedly crucial for the interpretation of the results. Furthermore, in certain parts of the results relating to mutants Y67E and Y67F, the results are given exclusively for Y67E, while the interpretation of the results for Y67F is omitted, thus losing the valuable comparative component of the experiment. Also, in the section showing the results for apoptosis, it would be useful to include a graphical representation of the distribution of cells by quadrant to provide additional visual interpretation of the data.
The discussion is not sufficiently related to the results. There is a lack of integration of the data obtained into a wider scientific context and a synthesis of the results in the form of a general conclusion that would summarise the contribution of the study and highlight its relevance. A final discussion summary is particularly important to ensure the clarity and completeness of the manuscript.
Finally, it should be noted that the references are mostly older than five years. Although this does not necessarily detract from the value of the work, it would be desirable to include more recent scientific findings that would make the work more up-to-date and relevant to current literature and research trends.
Author Response
- However, despite these contributions, the paper has a number of structural and substantive weaknesses that detract from its clarity and scientific basis. The main shortcoming of the paper is the lack of a clearly defined purpose and aim of the research. The authors do not mention specific hypotheses, or the potential translational or applied value of the results obtained. This makes it difficult for the reader to understand either the guiding principles of the research or the scientific motivation behind the work.
Response: As part of the introduction (second last paragraph) we have added a statement of the purpose of the study and the underlying hypothesis to be tested. We have also modified this paragraph to better explain the functional importance of Cytc phosphorylation by regulating the mitochondrial membrane potential and thus energy ad ROS production.
- The introductory part of the text is generally well written but needs some refinement. Firstly, the purpose and aim of the research are not clearly formulated. In addition, the last paragraph of the introduction contains data closer to the presentation of the results and their interpretation, which is inappropriate for the introduction from a methodological point of view; this information would be better included in the results and/or discussion section.
Response: As above, as part of the introduction (second last paragraph) we have added a statement of the purpose of the study and the underlying hypothesis to be tested. As in many papers, the last paragraph oftentimes “gives a taste” of the data presented and also provides an introduction to the novelty (i.e., new phosphorylation site) and the functional consequences of the specific topic (i.e., the role of residue Y67). To be responsive to the reviewer’s comment we have shorted this section and left the details for the results section.
- The methodological section contains several omissions that affect the transparency and reproducibility of the experiment. For example, the colouring of the gel with Coomassie blue is not stated, although it can be inferred from the context that it was used. Important technical data is also missing, such as the number of heart samples analysed, the manufacturers of the ion exchangers (DE52 – anionic and CM52 – cationic) and the manufacturer of the spectrophotometer used (Jasco V-570). This information is of crucial importance for the transparency and reproducibility of the analysis and should therefore be included.
Response: The Coomassie blue staining was previously stated in the methods in section “2.3. Mutagenesis, expression, and purification of recombinant Cytc”, in the results in section “3.2. Overexpression and purification of functional Cytc variants in E. coli cells”, and in the figure legend of Figure 2.
The Y67 phosphorylation of Cytc was identified in three bovine heart Cytc samples. This information has been added to the legend for Figure 1A.
The manufacturer of the ion exchange columns is Whatman. This information has been added to the methods section “2.1. Isolation of Cytc from bovine heart tissue”.
The manufacturer of the spectrophotometer is Jasco. This information has been added to the methods section “2.4. Concentration determination of Cytc”.
Another reviewer commented that the methods section should be significantly shortened. We have tried to balance that feedback (see reviewer 1 comment 1) with the feedback given here to increase the technical detail.
- The results section is detailed and informative but contains technical and content-related irregularities. The principle of certain methods is described several times unnecessarily, which should be omitted in order not to overload the text. Some descriptions of the results are repeated, e.g. ˝The next day the cells were exposed to either H2O2 (300 μM for 14 h) or staurosporine (1 μM for 5 h) ˝. Figure 2D is not mentioned in the text, although it is supposedly crucial for the interpretation of the results. Furthermore, in certain parts of the results relating to mutants Y67E and Y67F, the results are given exclusively for Y67E, while the interpretation of the results for Y67F is omitted, thus losing the valuable comparative component of the experiment.
Response: We agree that the discussion of the results mostly focuses on the phosphomimetic variant. To address this point, we have added a paragraph in which we discuss the effect of the Y67F replacement (second to last paragraph of the discussion section).
We have added the attribution for updated Figure 2E (previously Figure 2D) in the results in section “3.5. Redox potential, susceptibility to oxidation, and superoxide scavenging activity are decreased in phosphomimetic Cytc”.
- Also, in the section showing the results for apoptosis, it would be useful to include a graphical representation of the distribution of cells by quadrant to provide additional visual interpretation of the data.
Response: We have included the requested visualizations in Figure 6.
- The discussion is not sufficiently related to the results. There is a lack of integration of the data obtained into a wider scientific context and a synthesis of the results in the form of a general conclusion that would summarise the contribution of the study and highlight its relevance. A final discussion summary is particularly important to ensure the clarity and completeness of the manuscript.
Response: We thank the reviewer for pointing this out. We have added a conclusion section that summarizes the results and puts them into perspective and a wider scientific context. We believe that with all the revisions made, the manuscript is much better integrated, and the story is presented much more clearly.
- Finally, it should be noted that the references are mostly older than five years. Although this does not necessarily detract from the value of the work, it would be desirable to include more recent scientific findings that would make the work more up-to-date and relevant to current literature and research trends.
Response: We agree that the cited literature is mostly older than 2020 but the cited papers are also the most relevant ones. We added recently published paper from 2025 (Singh et al.), showing a new posttranslational modification (ubiquitination) of Cytc. We also included several other recent references that identified new functions of Cytc in the nucleus opening up a new field for Cytc research (Gonzalez-Arzola et al, 2022; Buzon et al., 2023; Casado-Combreras et al., 2024; and Tamargo-Azpilicueta et al., 2024).
Reviewer 3 Report
Comments and Suggestions for Authors
In this study by Wan and colleagues, the authors aim to advance the understanding of cytochrome c regulation through residue phosphorylation. Specifically, they extend their previous work by identifying additional phosphorylatable sites, using bovine heart as the initial experimental model. In this context, they identify another Y67, which they note is highly conserved across species and might be relevant for Cytc functions.
Through mutagenesis-based approaches in bacterial systems, they generate variants of Y67 that mimic that conserved the capacity of phosphorylation, as well as a non-phosphorylable control. Using these models, they demonstrate that the phosphorylated state of Y67 functionally restricts several roles of cytochrome c, particularly cardiolipin peroxidation, cellular respiration, mitochondrial membrane potential, antioxidant activity, and cell death.
Overall, the introduction is well constructed and provides a clear framework for understanding the manuscript. The data are clearly represented in the figures, and the manuscript includes multiple complementary and logically integrated techniques.
The significance of the findings is relatively high, with strong potential clinical relevance, rather than purely phenomenological interest.
as minor comments:
It would be helpful to identify and prioritize the kinases responsible for the phosphorylation of these residues, especially considering that their expression may vary across tissues, potentially explaining tissue-specific effects or disease associations.
Additionally, it may be worthwhile to explore which diseases or physiological conditions involve the phosphorylation of this residue and whether selective inhibition of the corresponding kinases could represent a viable therapeutic strategy.
Please revise the following sentence for clarity:
“which activates caspase-9 and the downstream caspase cascade-3”.
Latinisms such as in vivo, in vitro, etc., should be italicized.
The manuscript sometimes uses the three-letter code to designate amino acids (e.g., Met80), and in other instances, the one-letter code (e.g., Y67). Please standardize this throughout.
The following sentence may be misleading:
“Residue Y67 is highly conserved in mammals and other clades [41], further suggesting a functional relevance in Cytc regulatory roles for this phosphorylation”.
Conservation across species is not necessarily indicative of functional relevance, especially considering the authors themselves note in the introduction that
“Interestingly, the phosphorylations appear to be tissue-specific.”
This point should be acknowledged explicitly.
Figure 1: The text refers to Y67, while the figure shows Tyr67. Please harmonize the nomenclature.
Author Response
- It would be helpful to identify and prioritize the kinases responsible for the phosphorylation of these residues, especially considering that their expression may vary across tissues, potentially explaining tissue-specific effects or disease associations.
Response: We agree with the reviewer that the kinase and phosphatase as well as disease and physiological conditions involved in Y67 phosphorylation of Cytc are extremely important. While we cannot divulge too much at this time, we have identified both the kinase and phosphatase responsible for the phosphorylation of Cytc at Y67. Both the candidate kinase and phosphatase are highly implicated in a particular cancer. We are currently finishing a manuscript that covers the role of Cytc Y67 phosphorylation in this particular cancer. As you may note from the data seen here in this manuscript characterizing the Y67 phosphorylation, the modification seems to promote both inhibition of aerobic respiration and apoptosis evasion phenotypes, which are both hallmarks of cancer. We hope that you look forward to the follow-up manuscript soon.
- Additionally, it may be worthwhile to explore which diseases or physiological conditions involve the phosphorylation of this residue and whether selective inhibition of the corresponding kinases could represent a viable therapeutic strategy.
Response: See reviewer 3 comment 1.
- Please revise the following sentence for clarity:
“which activates caspase-9 and the downstream caspase cascade-3”.
Response: This sentence has been revised as requested.
- Latinisms such as in vivo, in vitro, etc., should be italicized.
Response: These changes have been made.
- The manuscript sometimes uses the three-letter code to designate amino acids (e.g., Met80), and in other instances, the one-letter code (e.g., Y67). Please standardize this throughout.
Response: All amino acids have been standardized to one-letter codes.
- The following sentence may be misleading:
“Residue Y67 is highly conserved in mammals and other clades [41], further suggesting a functional relevance in Cytc regulatory roles for this phosphorylation”.
Conservation across species is not necessarily indicative of functional relevance, especially considering the authors themselves note in the introduction that, “Interestingly, the phosphorylations appear to be tissue-specific.” This point should be acknowledged explicitly.
Response: We thank the reviewer for this comment, and we have borrowed the reviewer’s language to make this point clear at the beginning of the discussion section. We also added another reference in support of the statement that Y67 is a structurally important residue (Battistuzzi et al.). The reviewer is absolutely correct that the tissue-specificity is a separate point. In the beginning of the discussion section, we simply tried to make the argument that this residue is important for the structural integrity and function of Cytc and we did not wish to imply that it is conserved because it is a target for tissue-specific phosphorylation. We mentioned the quoted sentence in the introduction but not in the context of residue conservation.
- Figure 1: The text refers to Y67, while the figure shows Tyr67. Please harmonize the nomenclature.
Response: All amino acids have been standardized to one-letter codes.
Round 2
Reviewer 2 Report
Comments and Suggestions for Authors
The authors have taken the previous comments into account and responded to them constructively and specifically. In line with the suggestions, appropriate corrections have been made to the text, which have improved the clarity and scientific basis of the paper. The paper has been supplemented with relevant and up-to-date literature, and the purpose of the research has been further clarified, which has increased its scientific value. The introduction and discussion have been significantly expanded, making the contextualization of the topic clearer and more comprehensive. Also, the number of samples is now listed transparently and precisely, increasing methodological clarity. Overall, the content of the paper has been improved and has been given a new, higher quality dimension, which has significantly increased its suitability for publication.